# Diagnostic accuracy of the lumbar spinal stenosis-diagnosis support tool and the lumbar spinal stenosis-self-administered, self-reported history questionnaire

Ryoji Tominaga[1,2,3], Noriaki Kurita [3,4,5]*, Miho Sekiguchi[1], Koji Yonemoto[6,7], Tatsuyuki Kakuma[8], Shin-ichi Konno[1]

1 Department of Orthopaedic Surgery, Fukushima Medical University School of Medicine, Fukushima, Japan, 2 Department of Orthopaedic and Spinal Surgery, Aizu Medical Center, Fukushima Medical University, Fukushima, Japan, 3 Department of Clinical Epidemiology, Graduate School of Medicine, Fukushima Medical University, Fukushima, Japan, 4 Department of Innovative Research and Education for Clinicians and Trainees, Fukushima Medical University Hospital, Fukushima, Japan, 5 Center for Innovative Research for Communities and Clinical Excellence, Fukushima Medical University, Fukushima, Japan, 6 Division of Biostatistics, School of Health Sciences, Faculty of Medicine, University of the Ryukyus, Okinawa, Japan, 7 Advanced Medical Research Center, Faculty of Medicine, University of the Ryukyus, Okinawa Japan, 8 Biostatistics Center, Kurume University, Fukuoka, Japan

* kuritanoriaki@gmail.com

**Data Availability Statement:** All relevant data are within the article and its Supporting Information files.

## Abstract

Despite the applicability of the lumbar spinal stenosis (LSS)-diagnosis support tool (DST) and the LSS-self-administered, self-reported history questionnaire (SSHQ), their diagnostic accuracy has never been compared with that of the well-known North American Spine Society (NASS) clinical description of LSS. This study aimed to compare the diagnostic accuracy of the two diagnostic tools with that of the NASS guidelines' clinical description of LSS in a Japanese secondary care hospital setting. This multicenter cross-sectional study used data from the lumbar spinal stenosis diagnostic support tool (DISTO) project, which was conducted from December 1, 2011 to December 31, 2012. Japanese adults with low back pain (LBP) aged ≥20 years were consecutively included. The reference standard was LSS diagnosed by orthopedic physicians. The diagnostic accuracy of the two support tools was compared. Of 3,331 patients, 1,416 (42.5%) patients were diagnosed with LSS. The NASS clinical description of LSS had a sensitivity of 63.9% and specificity of 89.5%. The LSS-DST and LSS-SSHQ had sensitivities of 91.3% and 83.8% and specificities of 76.0% and 57.6%, respectively, with substantial improvements in sensitivity ($P < 0.0001$). Similar results were obtained when we limited included patients to those aged >60 years. These findings indicated that the LSS-DST and LSS-SSHQ were more sensitive in screening patients with LBP for a diagnosis of LSS than the NASS clinical description of LSS. This study strongly supports prioritizing the use of either of these two diagnostic support tools for screening.

**Funding:** The authors received no specific funding for this work.

**Competing interests:** The authors have declared that no competing interests exist.

# 1. Introduction

Lumbar spinal stenosis (LSS) is a common musculoskeletal disorder in the aging population, with a prevalence rate of approximately 11% in the general population [1]. An accurate diagnosis of LSS is challenging due to a lack of consensus concerning definitive diagnostic criteria and the requirement for consistency between physical manifestations and imaging features. Specifically, expert clinicians should diagnose LSS through careful physical examinations and consistent findings in imaging examinations, including roentgenography, computed tomography (CT), and magnetic resonance imaging (MRI). To facilitate this challenging diagnosis, numerous clinical definitions and diagnostic support tools for LSS have been developed [2–8]. Nonetheless, the diagnostic performance of these diagnostic aids has not yet been fully compared.

Two diagnostic support tools, the LSS-diagnosis support tool (LSS-DST) and the LSS-self-administered self-reported history questionnaire (LSS-SSHQ), have been developed in Japan to aid primary care physicians in accurately identifying patients with LSS and to provide appropriate care [3, 4]. The LSS-DST and LSS-SSHQ have been rated as having level II diagnostic evidence for LSS by the Degenerative LSS Work Group of the North American Spine Society (NASS) Evidence-Based Clinical Guideline Development Committee [2]. In addition, these support tools have shown good applicability according to the latest systematic review that used the Quality Assessment of Diagnostic Accuracy Studies (QUADAS)-2 assessment tool [9].

A new cut-off value for the diagnostic accuracy of the LSS-SSHQ in primary care settings has been reported [10]. However, diagnostic accuracy measures derived from research studies may not reflect real-world properties due to a lack of external validation and the possibility of a unideal diagnostic flow. Indeed, in that study, a definitive diagnosis of LSS was only partially guided by the LSS-DST and false negatives (i.e., missed diagnoses) may have occurred. The clinical description of LSS found in the NASS guidelines is the most common reference [2], and primary care physicians or orthopedic residents may utilize this clinical description when examining a patient with suspected LSS. However, it remains unclear how accurately this clinical description helps to identify LSS. Therefore, the superiority of the aforementioned two diagnostic support tools over the NASS diagnostic guidelines must be externally validated for situations in which a definitive diagnosis is made solely by an orthopedic surgeon.

This large-scale, multicenter, cross-sectional study aimed to compare the diagnostic test accuracies between the two support tools and the clinical description of LSS in the NASS diagnostic guidelines at secondary care hospitals in Japan. We hypothesized that the two diagnostic support tools for LSS would be more sensitive and more useful for screening than the clinical description of LSS in the NASS diagnostic guidelines.

# 2. Materials and methods

## 2.1 Study design and data collection

This multicenter cross-sectional study used data from the Lumbar Spinal Stenosis Diagnostic Support Tool (DISTO) project, which was conducted from December 1, 2011 to December 31, 2012. The DISTO project was implemented in 1657 medical institutions under the guidance of the Japanese Society for Spine Surgery and Related Research (JSSR) to verify awareness and the diagnostic accuracy of a lumbar spinal stenosis diagnostic support tool in order to contribute to early detection and treatment of LSS. Recruitment for study participation was announced on the JSSR website, and the study was conducted at facilities that expressed a willingness to participate. An LSS-DST checklist and the NASS clinical description of LSS were

distributed to participating medical facilities. The physician-in-charge completed the checklist, in addition to providing usual medical care. Patients who agreed to participate in the study were asked to complete the LSS-SSHQ prior to their consultation. The DISTO project collected and analyzed the checklist and the diagnostic information provided by the physician concerning LSS, peripheral artery disease (PAD), and diabetes mellitus (DM). The target population included patients with low back pain (LBP) aged ≥20 years who had undergone a medical examination, irrespective of the reason for visiting secondary care hospitals with an orthopedic department. Patients were included only based on their LBP symptom, regardless of leg symptoms or the duration of LBP. Participants were consecutively recruited from December 1, 2011, to December 31, 2012. Exclusion criteria comprised patients with the following: heart failure, renal failure, respiratory failure, hepatic failure, a decreased level of consciousness, a history of psychiatric disorders (e.g., schizophrenia or personality disorders), and a history of spinal surgery. The ethics committees of Fukushima Medical University (No. 1136) and the Japanese Orthopaedic Association approved this study. The participants were informed that data from the study would be submitted for publication and they provided their written informed consent.

## 2.2 Reference standards for LSS

The reference standard for LSS was a final diagnosis of LSS by the orthopedic physicians in charge of the participants. Participants were carefully assessed based on their medical history, the results of a detailed physical examination, and radiological findings from modalities such as radiography, CT, and MRI. In the absence of universally acceptable diagnostic criteria for LSS, decision-making by a professional clinician was adopted to establish an accurate diagnosis.

## 2.3 Index tests

**2.3.1 The LSS-DST.** The LSS-DST is a brief clinical diagnostic tool that helps physicians precisely diagnose patients with LSS (**Table 1**) [3]. It consists of 10 items that are grouped into

**Table 1. A clinical DST for identifying patients with LSS (LSS-DST).**

| Items | Score |
|---|---|
| History | |
| Age (years) | |
| 60–70 | +1 |
| >70 | +2 |
| Absence of diabetes mellitus | +1 |
| Symptom | |
| Intermittent claudication, positive | +3 |
| Exacerbation of symptoms when standing up | +2 |
| Symptom improvement when bending forward | +3 |
| Physical examination | |
| Symptoms induced by having patients bend forward | -1 |
| Symptoms induced by having patients bend backwards | +1 |
| Ankle brachial index ≥0.9 | +3 |
| Abnormal Achilles tendon reflex | +1 |
| SLR test result, positive | -2 |

If the total score is >7, there is a high possibility of LSS.

DST, diagnosis support tool; LSS, lumbar spinal stenosis; SLR, straight leg raising.

three main categories, namely, medical history, symptoms, and physical examination. The LSS-DST can be scored by primary care physicians within their usual resources without the need for special equipment or imaging studies. There is a positivity cut-off point of 7, where the area under the receiver operating characteristic (ROC) curve is the highest. At this cut-off point of 7, the sensitivity and specificity of the LSS-DST have been reported to be 92.8% and 72.0%, respectively [3]. All participating orthopedic physicians consented to use the LSS-DST for each patient.

**2.3.2 The LSS-SSHQ.** The LSS-SSHQ was developed to evaluate the diagnostic value of the medical history of patients with LSS (**Table 2**) [4]. This self-completed questionnaire comprises 10 items concerning subjective symptoms only. The LSS-SSHQ can be distributed to patients by primary care physicians unfamiliar with neurological physical examination. Scoring can be completed by the patients or their primary care physicians. One validation study reported a sensitivity and specificity of 84% and 78%, respectively, with an area under the ROC curve of 0.782 [4]. We adopted a new cut-off point for the LSS-SSHQ (LSS-SSHQ version 1.1; a total score of 3 on Q1–Q4 or a score of ≥1 on Q1–Q4 and ≥2 on Q5–Q10 indicated positivity), as this cut-off point had higher sensitivity and negative predictive value (NPV) than the original value used in primary care settings [10]. All patients completed the LSS-SSHQ, to which version 1.1 scoring was later applied. The written questionnaires were collected so that the attending physician could not refer to them.

## 2.4 Typical clinical descriptions of LSS according to the NASS

As one of the most universally recognized presentations of LSS, the following clinical descriptions written in the NASS guidelines were used in the present study [2]:

"1. Degenerative LSS describes a condition in which there is diminished space available for the neural and vascular elements in the lumbar spine secondary to degenerative changes in the spinal canal.

2. When symptomatic, this causes a variable clinical syndrome of gluteal and/or lower-extremity pain and/or fatigue, which may occur with or without back pain.

3. Provocative features include upright exercise, such as walking or positionally induced neurogenic claudication. Palliative features commonly include symptomatic relief with forward flexion, sitting, and/or recumbency."

**Table 2. A self-administered, self-reported history questionnaire for identifying patients with LSS (LSS-SSHQ).**

| Items |
| --- |
| Q1: Numbness and/or pain in the thighs down to the calves and shins |
| Q2: Numbness and/or pain increase in intensity after walking for a while, but relieved through taking a rest |
| Q3: Standing for a while brings on numbness and/or pain in the thighs down to the calves and shins |
| Q4: Numbness and/or pain are reduced by bending forward |
| Q5: Numbness is present in both legs |
| Q6: Numbness is present in the soles of both feet |
| Q7: Numbness arises around the buttocks |
| Q8: Numbness is present, but pain is absent |
| Q9: A burning sensation arises around the buttocks |
| Q10: Walking nearly causes urination |

A total score of 3 on Q1–Q4 or a score ≥1 on Q1–Q4 and a score of ≥2 on Q5–Q10 indicated the presence of LSS. LSS, lumbar spinal stenosis; SSHQ, self-administered, self-reported history questionnaire.

Given that the description in point 1 is a morphological feature and not testable in a clinical practice setting, we adopted the descriptions set out in points 2 and 3 for LSS and considered them to represent typical clinical presentations in this study. Attending physicians assessed patients based on these descriptions using a checklist.

## 2.5 Statistical analyses

Demographic characteristics, comorbidities, and outcomes were analyzed using descriptive statistics. To evaluate the diagnostic test accuracy of the clinical description of LSS in the NASS diagnostic guidelines, the LSS-DST, and the LSS-SSHQ, the sensitivity and specificity of each index test was examined. In addition, the sensitivities and specificities of the LSS-DST and LSS-SSHQ were compared with those of the clinical description of LSS in the NASS guidelines using the McNemar test [11]. Furthermore, the NPVs of the three tools were also calculated, as it is important to determine the number of false positives obtained by physicians who were unskilled in examining LSS when using these tools clinically. To examine the overall diagnostic accuracy of the three index tests, we also calculated the diagnostic odds ratio (DOR) according to the following equation: DOR = (sensitivity × specificity)/ (1-sensitivity × 1-specificity) [12].

Several sensitivity analyses were performed. First, we included participants with a total score of >7 and those with a score of ≤3, despite lacking ankle brachial index (ABI) and other values, and these scores were regarded as being positive or negative for the LSS-DST, respectively. Second, we performed a sensitivity analysis limited to those aged ≥60 years. All statistical analyses were performed using SAS version 9.3 (SAS Institute Inc., Cary, NC, USA) software. A $P$-value <0.05 was considered to indicate statistical significance.

## 3. Results

### 3.1 Patient background

Overall, 10,669 patients with LBP participated in this study. After excluding 7,338 patients with missing or inappropriate data in relation to the LSS-DST and LSS-SSHQ, 3,331 participants were included in the primary analysis (**Fig 1**, **S1 Fig**). Numerous participants (n = 4,082) did not undergo an ABI assessment, as the ABI was usually only performed for patients with suspected PAD. **Table 3** presents the study participants' characteristics. In total, 1,755 men and 1,564 women (12 cases of missing sex data) were examined by hospital-based orthopedists.

### 3.2 Outcome data

LSS was prevalent in 42.5% of the population. Test results obtained using the LSS-DST, LSS-SSHQ, and the NASS clinical description of LSS are shown in **Table 4**. Only 63.9% of patients with LSS met the NASS clinical description of LSS (sensitivity 63.9% [95% confidence interval (CI) 61.4%–66.4%]), while 89.5% of patients without LSS did not meet this condition (specificity 89.5% [95% CI 88.1%–90.9%]) (**Table 4**).

The sensitivity of the LSS-DST was superior to that of the NASS clinical description (91.3% [95% CI 89.9%–92.8%] vs. 63.9% [95% CI 61.4%–66.4%], $P$ < 0.0001; **Table 4**); however, its specificity was inferior to that of the NASS clinical description (76.0% [95% CI 74.1%–77.9%] vs. 89.5% [95% CI 88.1%–90.9%], $P$ < 0.0001; **Table 4**).

The LSS-SSHQ also exhibited superior sensitivity when compared with the NASS clinical description (83.8% [95% CI 81.8%–85.7%] vs. 63.9% [95% CI 61.4%–66.4%], $P$ < 0.0001; **Table 4**). However, the specificity of the LSS-SSHQ was inferior to that of the NASS clinical

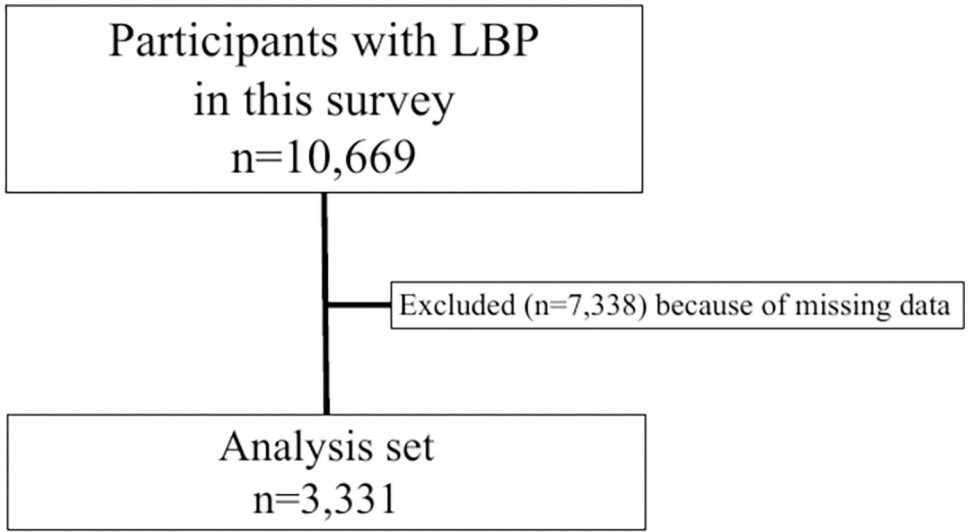

**Fig 1. Flow chart of participant inclusion.** LBP, low back pain.

description (57.6% [95% CI 55.3%–59.8%] vs. 89.5% [95% CI 88.1%–90.9%], $P < 0.0001$; **Table 4**).

The NPVs were 0.77 (95% CI 0.75–0.79) for the NASS clinical description, 0.92 (95% CI 0.91–0.94) for the LSS-DST, and 0.83 (95% CI 0.81–0.85) for the LSS-SSHQ (**Table 5**).

The DORs for each index test were 15.1 (95% CI 12.6–18.1) for the NASS clinical description, 33.3 (95% CI 26.9–41.1) for the LSS-DST, and 7.0 (95% CI 5.9–8.3) for the LSS-SSHQ (**Table 6**).

**Table 3. Patient characteristics.**

| Characteristic | n (%) | Missing data |
|---|---|---|
| Age (years) | | - |
| 20–29 | 166 (5.0) | |
| 30–39 | 310 (9.3) | |
| 40–49 | 308 (9.3) | |
| 50–59 | 411 (12.3) | |
| 60–69 | 892 (26.8) | |
| 70+ | 1244 (37.3) | |
| Sex | | 12 |
| Male | 1755 (52.9) | |
| Female | 1564 (47.1) | |
| Presence of LSS | | - |
| (-) | 1915 (57.5) | |
| (+) | 1416 (42.5) | |
| Presence of DM | | - |
| (-) | 3012 (90.4) | |
| (+) | 319 (9.6) | |
| Presence of PAD | | - |
| (-) | 2256 (67.7) | |
| (+) | 1075 (32.3) | |

DM, diabetes mellitus; LSS, lumbar spinal stenosis; PAD, peripheral arterial disease; (-), not present; (+), present.

**Table 4. Sensitivity and specificity of the NASS clinical description of LSS, LSS-DST, and LSS-SSHQ.**

| Index test | Sensitivity | | P-value for heterogeneity | Specificity | | P-value for heterogeneity |
|---|---|---|---|---|---|---|
| | Point estimate | 95% CI | | Point estimate | 95% CI | |
| 1) NASS clinical description of LSS | 63.9% | 61.4%–66.4% | | 89.5% | 88.1%–90.9% | |
| 2) LSS-DST | 91.3% | 89.9%–92.8% | 2) vs. 1) <0.0001 | 76.0% | 74.1%–77.9% | 2) vs. 1) <0.0001 |
| 3) LSS-SSHQ | 83.8% | 81.8%–85.7% | 3) vs. 1) <0.0001 | 57.6% | 55.3%–59.8% | 3) vs. 1) <0.0001 |

CI, confidence interval; DST, diagnosis support tool; LSS, lumbar spinal stenosis; NASS, North American Spine Society; SSHQ, self-administered, self-reported history questionnaire.

Similar results were obtained when patients (n = 7,914) with >7 points on the LSS-DST without ABI data were treated as LSS-DST-positive and patients with <7 points without ABI data were treated as LSS-DST-negative (**S1 and S2 Tables**). When patients aged >60 years (n = 2,136) were included in the sensitivity analysis, the results were similar to those of the main analysis (**S3 and S4 Tables).**

## 4. Discussion

In this study, which was conducted in secondary care hospital settings, the LSS-DST and LSS-SSHQ had significantly higher sensitivity for diagnosing LSS in patients with LBP than the clinical description of LSS in the NASS diagnostic guidelines.

Several diagnostic support tools for LSS have been developed, and each of these tools has been reported to have high sensitivity. The LSS-DST, which was the first diagnostic support tool used for LSS, was developed for patients aged >20 years with primary symptoms of pain or numbness in the legs in a Japanese hospital setting, including university hospitals, medical centers, and clinics affiliated with such hospitals. This 10-item tool (two items for medical history, three items for patients' symptoms, and five items for physical examination) has been reported to have a sensitivity of 92.8% and a specificity of 72.0% [3]. The LSS-SSHQ was then later developed to assess the diagnostic value of a patient's history in a hospital setting. The sensitivity and specificity of the LSS-SSHQ have been reported to be 84% and 78%, respectively [4]. The LSS-SSHQ was externally validated in a Japanese primary care setting, and its sensitivity has improved with the introduction of a new cut-off value (79.8% vs. 68.3%) [10]. However, no studies have compared the diagnostic accuracy of these diagnostic support tools among identical participants despite the influence of the patient spectrum (i.e., variations in the severity of the targeted disease and differential diagnosis) on sensitivity. To our knowledge, this is the first study to compare the diagnostic accuracy of these diagnostic support tools and the clinical description noted in the NASS guidelines for patients suspected of having LSS in the same patient spectrum.

**Table 5. NPVs for the NASS clinical description of LSS, LSS-DST, and LSS-SSHQ.**

| Index test | NPVs | |
|---|---|---|
| | Point estimate | 95% CI |
| 1) NASS clinical description of LSS | 0.77 | 0.75–0.79 |
| 2) LSS-DST | 0.92 | 0.91–0.94 |
| 3) LSS-SSHQ | 0.83 | 0.81–0.85 |

CI, confidence interval; DST, diagnosis support tool; LSS, lumbar spinal stenosis; NASS, North American Spine Society; NPVs, negative predictive values; SSHQ, self-administered, self-reported history questionnaire.

**Table 6. DORs of the NASS clinical description of LSS, the LSS-DST, and the LSS-SSHQ.**

| Index test | DOR | |
|---|---|---|
| | Point estimate | 95% CI |
| 1. NASS clinical description of LSS | 15.1 | 12.6–18.1 |
| 2. LSS-DST | 33.3 | 26.9–41.1 |
| 3. LSS-SSHQ | 7.0 | 5.9–8.3 |

CI, confidence interval; DORs, diagnostic odds ratios; DST, diagnosis support tool; LSS, lumbar spinal stenosis; NASS, North American Spine Society; SSHQ, self-administered, self-reported history questionnaire.

We consider that the findings in this study may influence the activities of both physicians and epidemiological researchers for several reasons. First, the improved sensitivity of the two diagnostic support tools when compared with the clinical description of LSS in the NASS diagnostic guidelines provides evidence in support of their use in screening for the diagnosis of LSS. The high NPVs of the diagnostic support tools for LSS are important because of the clinical significance in terms of effectively limiting false-negative results, even for clinicians unfamiliar with diagnosing and treating patients with LSS. These two support tools can be useful, especially for primary care physicians and junior orthopedic surgeons, in conjunction with the universally recognized clinical description of LSS in the NASS diagnostic guidelines [2]. Further, these tools have been referred to in the management of patients with LBP and/or lower-extremity symptoms. A correct diagnosis of LSS is often difficult to achieve, particularly in the early stages, and extraspinal disorders such as PAD, diabetes-related peripheral neuropathy (DPN), and other musculoskeletal diseases can be misdiagnosed [13, 14]. The superior sensitivity of the two support tools can be explained by additional factors that are considered for differential diagnosis in actual practice, including age, the presence of DM, the ABI, and the results of a straight leg raising test. In addition, clinicians also consider lower extremity symptoms, intermittent claudication, and postural factors described in the NASS guidelines.

Second, the excellent sensitivity of these diagnostic support tools is likely to reduce unnecessary and costly imaging tests such as CT and MRI scans. The high sensitivity of these two tools and their low cost may also facilitate large-scale epidemiological studies on LSS. To date, data on the population-based epidemiology of LSS are relatively limited. This is due, in part, to the difficulty in diagnosing LSS [15] given the absence of objective diagnostic criteria, even with the use of imaging tests [16]. In this study, the LSS-DST had the highest DOR of the three tools, and the estimated DOR for the LSS-DST in detecting LSS was 33.3. This suggests that, when using the LSS-DST, the odds of positivity among patients with LSS is 33.3 times higher than the odds of positivity among patients without LSS. The DOR is a simple and statistically tractable indicator that can be used to assess diagnostic accuracy without the need for other indicators [12], and the results of this study indicated the LSS-DST was one of the best available screening methods for LSS. Moreover, this study verified the external validity of the diagnostic accuracy of the two diagnostic support tools, which have been considered to have good applicability according to the QUADAS-2 assessment tool [9].

This study had several strengths. First, the large-scale nationwide study design and inclusion of >3,000 participants ensured the generalizability of our findings regarding the usefulness of the two diagnostic support tools. Second, the use of diagnosis by orthopedic surgeons as a reference standard reflects the best current diagnostic practice. Although the large number of facilities participating in the study may make standardization of clinicians' diagnostic procedures challenging, we consider that we were able to estimate the diagnostic accuracies of the screening tools at current standards of medical care. Third, similar results were obtained

through sensitivity analysis conducted on two different populations, indicating that the detected results were robust.

Nevertheless, this study also had several limitations. First, no expert consensus has been reached regarding the reference standard for diagnosing LSS. According to a recent study, expert consensus building is recommended when conducting research on diagnostic accuracy for diseases for which a clear diagnostic definition has not been established [17]. However, it was impractical to build expert consensus on LSS for each participant, as this was a multicenter nationwide study involving thousands of patients. Second, some of our findings obtained in outpatient settings at orthopedic hospitals may not be applicable to primary care settings. As more patients are likely to have severe LSS in this setting than in the primary care setting, the sensitivity of the three index tests (LSS-DST, LSS-SSHQ, and the clinical description in the NASS guidelines) may be higher in secondary care settings than in primary care settings (i.e., spectrum bias). Further studies are warranted to confirm whether differences in diagnostic accuracy between the two diagnostic support tools and the NASS clinical descriptions esti- mated in this study apply to primary care settings. Third, as the physician who utilized the LSS-DST and the NASS description was also involved in the determination of the reference standard, the diagnosis of LSS may have been guided by the results of the LSS-DST [18]. Therefore, the sensitivities obtained in the current study may be higher than the actual sensi- tivities. Fourth, we did not have data on the severity of diseases that were used as exclusion cri- teria. It is possible that several patients with diseases such as DPN and PAD were excluded, and this should be considered when interpreting the results of this study. Fifth, there is the pos- sibility of selection bias because many participants were not included in the main analysis due to missing values. Therefore, we compared participants included in the main analysis with those who had earlier been excluded (**S5 Table**). Participants included in the main analysis were older, there was a higher proportion of men than women, and there was a higher pres- ence of LSS. Therefore, this may suggest that selection bias may be introduced in the partici- pants included in the study. However, we included participants with a total score of >7 and those with a score of ≤3, despite lacking ABI and other values in sensitivity analysis. The num- ber of participants in the sensitivity analysis was increased to 7,914, and the results were similar to those of the main analysis. Therefore, we consider the results of our study were robust.

## 5. Conclusion

The LSS-DST and LSS-SSHQ were significantly more sensitive than the clinical description of LSS in the NASS diagnostic guidelines, based on our analysis of data from a large population of patients with LBP. Prioritizing the use of either of these two diagnostic support tools for screening should be emphasized in clinical practice.

## Supporting information

**S1 Fig. Details of excluded cases because of missing data.** ABI, ankle brachial index; DST, diagnosis support tool; LSS, lumbar spinal stenosis; NASS, North American Spine Society; SSHQ, self-administered, self-reported history questionnaire.
(DOCX)

**S1 Table. Sensitivity and specificity of the NASS clinical description of LSS, LSS-DST, and LSS-SSHQ.** In this analysis, participants with >7 points on the LSS-DST, despite missing ABI and other values, were treated as LSS-DST-positive, and participants with <7 points, despite missing ABI and other values, were treated as LSS-DST-negative (n = 7,914). ABI, ankle bra- chial index; CI, confidence interval; DST, diagnosis support tool; LSS, lumbar spinal stenosis;

NASS, North American Spine Society; SSHQ, self-administered, self-reported history questionnaire.
(DOCX)

**S2 Table. DORs of the NASS clinical description of LSS, LSS-DST, and LSS-SSHQ.** In this analysis, participants with >7 points on the LSS-DST, despite missing ABI and other values, were treated as LSS-DST-positive, and participants with <7 points, despite missing ABI and other values, were treated as LSS-DST-negative (n = 7,914). ABI, ankle brachial index; CI, confidence interval; DORs, diagnostic odds ratios; DST, diagnosis support tool; LSS, lumbar spinal stenosis; NASS, North American Spine Society; SSHQ, self-administered, self-reported history questionnaire.
(DOCX)

**S3 Table. Sensitivity and specificity of the NASS clinical description of LSS, LSS-DST, and LSS-SSHQ in participants aged >60 years (n = 2,136).** CI, confidence interval; DST, diagnosis support tool; LSS, lumbar spinal stenosis; NASS, North American Spine Society; SSHQ, self-administered, self-reported history questionnaire.
(DOCX)

**S4 Table. DORs of the NASS clinical description of LSS, LSS-DST, and LSS-SSHQ in participants aged >60 years (n = 2,136).** CI, confidence interval; DORs, diagnostic odds ratios; DST, diagnosis support tool; LSS, lumbar spinal stenosis; NASS, North American Spine Society; SSHQ, self-administered, self-reported history questionnaire.
(DOCX)

**S5 Table. Comparison of the participants who were included and those who were excluded.** Each comparison was subjected to a χ-square test. LSS, lumbar spinal stenosis.
(DOCX)

## Acknowledgments

The authors thank the participants of the DISTO project. This project was conducted under the supervision of the Japanese Society for Spine Surgery and Related Research. The authors also thank Editage (www.editage.com) for English language editing.

## Author Contributions

**Conceptualization:** Ryoji Tominaga, Miho Sekiguchi, Tatsuyuki Kakuma, Shin-ichi Konno.

**Data curation:** Koji Yonemoto.

**Formal analysis:** Noriaki Kurita, Koji Yonemoto.

**Funding acquisition:** Miho Sekiguchi, Tatsuyuki Kakuma, Shin-ichi Konno.

**Investigation:** Miho Sekiguchi, Shin-ichi Konno.

**Methodology:** Ryoji Tominaga, Noriaki Kurita, Koji Yonemoto.

**Project administration:** Ryoji Tominaga, Koji Yonemoto, Shin-ichi Konno.

**Resources:** Miho Sekiguchi, Shin-ichi Konno.

**Supervision:** Noriaki Kurita, Miho Sekiguchi, Shin-ichi Konno.

**Visualization:** Ryoji Tominaga.

**Writing – original draft:** Ryoji Tominaga, Noriaki Kurita.

**Writing – review & editing:** Miho Sekiguchi, Tatsuyuki Kakuma, Shin-ichi Konno.

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
