## [Decision Letter · Decision Letter 0]

28 Feb 2022

PONE-D-21-38525Diagnostic accuracy of the Lumbar Spinal Stenosis-Diagnosis Support Tool and the Lumbar Spinal Stenosis-Self-administered, Self-reported History QuestionnairePLOS ONE

Dear Dr. Kurita,

Thank you for submitting your manuscript to PLOS ONE. After careful consideration, we feel that it has merit but does not fully meet PLOS ONE’s publication criteria as it currently stands. Therefore, we invite you to submit a revised version of the manuscript that addresses the points raised during the review process.

Please amend the manuscript according to the suggestion of the reviewers. Please discuss the reasons, where this might not be possible.

We look forward to receiving your revised manuscript.

Kind regards,

Michael C Burger, M.D.

Academic Editor

PLOS ONE

Journal Requirements:

2. Please provide additional details regarding participant consent. In the ethics statement in the Methods and online submission information, please ensure that you have specified (1) whether consent was informed and (2) what type you obtained (for instance, written or verbal, and if verbal, how it was documented and witnessed). If your study included minors, state whether you obtained consent from parents or guardians. If the need for consent was waived by the ethics committee, please include this information

Reviewers' comments:

Reviewer's Responses to Questions

**Comments to the Author**

1. Is the manuscript technically sound, and do the data support the conclusions?

Reviewer #1: Yes

Reviewer #2: Partly

Reviewer #3: Yes

Reviewer #4: Yes

2. Has the statistical analysis been performed appropriately and rigorously? 

Reviewer #1: Yes

Reviewer #2: Yes

Reviewer #3: Yes

Reviewer #4: Yes

3. Have the authors made all data underlying the findings in their manuscript fully available?

Reviewer #1: Yes

Reviewer #2: Yes

Reviewer #3: Yes

Reviewer #4: Yes

4. Is the manuscript presented in an intelligible fashion and written in standard English?

Reviewer #1: Yes

Reviewer #2: Yes

Reviewer #3: Yes

Reviewer #4: Yes

5. Review Comments to the Author

Reviewer #1: I congratulate the authors on a well conducted study and well-written manuscript. I have only minor suggestions for revisions and some areas for consideration. I am happy to review a revised version of the manuscript.

Introduction

Line 2. Suggest to include prevalence estimates from Jensen 2021 “Prevalence of lumbar spinal stenosis in general and clinical populations: a systematic review and meta-analysis” (10.1007/s00586-020-06339-1) as this is the most recent and comprehensive review of prevalence estimates for LSS.

Lines 56-58. I question whether the diagnosis needs to be made by an orthopedic surgeon. There are many other medical specialties (and health professionals) that are primary contact for people with LSS. In fact, on line 76 you report that primary care physicians also diagnose LSS. At minimum, I would suggest a reference for the statement that LSS need be diagnosed by a well-trained orthopedic surgeon.

Line 66. Suggest to define NASS for readers here and remove from line 75.

Line 80. Suggest to add “diagnostic guidelines” after NASS.

Materials and methods

Line 90. I would appreciate more clarity on how this study data was collected. 1,657 hospitals is a large recruitment procedure. Was this data collected specifically for this study, or integrated into the standard data collection across these centers? Perhaps a national registry?

Results

Line 181. Were those not included different from those that were? Is the data available to check this? It is important to know how these individuals differed in clinical characteristics. A selection bias may have been introduced where people with more severe disease were included, therefore inflating the diagnostic accuracy of the tools. For example, it appears that the age, sex, and presence of LSS could be compared between those included and those with missing or inappropriate data (n=7,338).

Figure 1. It is not clear how participants were excluded from the figure. I suggest that the “Excluded box” contain only those with missing data that were excluded. As written now, the numbers listed under exclusion sum to much greater than 7338. Please report missing data in a separate box. It would also be helpful to report how many patients were screened but declined to participate.

Line 183. It would be helpful to have more baseline characteristic information on all participants, since the diagnostic accuracy of a tool is dependent on the study population. Perhaps there was no other data collected, but if so, I would highly recommend greater description of the study cohort.

Discussion

Lines 284-286. I am not convinced that real world diagnosis should be considered a strong gold-standard/criterion measure. I do agree with the authors in that specialist diagnosis is the best that is currently available, but I would like to see some discussion around the fact that a clinician-based reference standard test can be problematic. In fact, the authors do later discuss the limitations of having no consensus diagnostic definition for LSS, so I would suggest that the language around “our findings on diagnostic accuracy can be considered valid” be softened and some additional uncertainty be presented. It would also be prudent to comment on the potential variation in clinician diagnosis standards across the large number of participating clinicians. It is probable that different clinicians would arrive at conflicting results on the reference standard for LSS, at least in a proportion of patients.

Line 299-301. It is a bold claim that the care setting is unlikely to alter the comparison of the three index tests. I do, however, respect the right of the authors to make this claim, but suggest that some reasoning be provided. Why would spectrum bias not be a concern here?

Reviewer #2: This study was conducted to compare the sensitivity and specificity of lumbar spinal stenosis diagnosis utilizing the Lumbar Spinal Stenosis-Diagnosis Support Tool (LSS-DST), the Lumbar Spinal Stenosis - Self-administered, Self-reported History Questionnaire (LSS-SSHQ) and the clinical description of LSS from the NASS diagnostic guidelines in a secondary care hospital setting. The authors concluded that “the LSS-DST and LSS-SSHQ had significantly higher sensitivity for diagnosing LSS in patients with LBP than the clinical description of LSS from the NASS diagnostic guidelines.” This reviewer has some criticisms, which should be addressed by the authors.

These tools have been used to “support” the diagnosis of LSS. This reviewer thinks that the most important thing in the management patients with LSS does prevent mis diagnosis when physicians, who are not spine specialists, use these tools in clinical setting. Considering the clinical relevancy, this reviewer thinks that it is important to precise if the score was not involved in the setting, the diagnosis of patients, who examined by the tools, must be “not” LSS. Therefore, the negative predictive value of these tools is more important than the sensitivity and specificity. Negative predictive values should be shown and compared among three tools.

Utilizing the typical clinical description of LSS according to the North American Spine Society (NASS), this reviewer does think that the description in point 1 is important items to diagnose the LSS. If this description was excluded, the NASS description to diagnose the LSS should not be used in the present study.

Reviewer #3: This paper is well written.

This study is a large-scale cross-sectional study and is noteworthy.

We agree that the two tools used in this study, LSS-DST and LSS-SSHQ can be used to improve the diagnostic accuracy of LSS.

Reviewer #4: Q1. Why do authors hypothesize that LSS-DST and LSS-SSHQ would be more sensitive and more useful for screening than the clinical description of LSS from the NASS diagnostic guidelines? (Line 85-86)

Q2. Did the author use the list or questionnaire to confirm the clinical description of LSS from the NASS diagnostic guidelines? Or did the authors confirm the clinical description of LSS from the NASS diagnostic guidelines from the items of LSS-DST and LSS-SSHQ? Because at least from the previous reports from your group, the reviewer could not find the description regarding “the clinical description of LSS from the NASS diagnostic guidelines”.

Comment:

1. Authors emphasized sensitivity or screening for LSS. But authors also discuss the specificity, the reviewer thinks.

2. I clinical setting, how do primary care physician use LSS-DST, LSS-SSHQ and the clinical description of LSS from the NASS diagnostic guidelines. Authors should describe it.

6. PLOS authors have the option to publish the peer review history of their article (what does this mean?). If published, this will include your full peer review and any attached files.

Reviewer #1: **Yes: **James J. Young

Reviewer #2: No

Reviewer #3: No

Reviewer #4: No

---

## [Author Response · Author response to Decision Letter 0]

13 Apr 2022

Response to the Comments 

We wish to express our gratitude to the Reviewers for their insightful comments that have helped us to significantly improve our paper. We have revised our manuscript and tables according to the feedback we received. Please check our revised files as well as our responses to your comments below.

Responses to Editor

Reviewer #1

Line 2. Suggest to include prevalence estimates from Jensen 2021 “Prevalence of lumbar spinal stenosis in general and clinical populations: a systematic review and meta-analysis” (10.1007/s00586-020-06339-1) as this is the most recent and comprehensive review of prevalence estimates for LSS.

Response: Thank you for pointing this out. Indeed, the review you suggested provides the most recent and comprehensive review prevalence estimates for LSS. We have therefore revised the relevant sentence as shown below. The references cited have also been revised accordingly.

“Lumbar spinal stenosis (LSS) is a common musculoskeletal disorder in the aging population, with a prevalence rate of approximately 11% in the general population [1].” [Page 5, Lines 56]

Lines 56-58. I question whether the diagnosis needs to be made by an orthopedic surgeon. There are many other medical specialties (and health professionals) that are primary contact for people with LSS. In fact, on line 76 you report that primary care physicians also diagnose LSS. At minimum, I would suggest a reference for the statement that LSS need be diagnosed by a well-trained orthopedic surgeon.

Response: Thank you for your meaningful remarks. In our previous studies, diagnosis by an orthopedic surgeon skilled in examining LSS has been used as the reference standard (i.e., the gold standard) to examine diagnostic accuracy [1-3]. However, as you point out, there are many medical specialties involved in LSS in clinical settings and, unlike in research, diagnosis by a skilled orthopedic surgeon is not essential for the diagnosis of LSS. Therefore, we have made the following modification in the relevant text.

“Specifically, expert clinicians should diagnose LSS through careful physical examinations and consistent findings in imaging examinations, including roentgenography, computed tomography (CT), and magnetic resonance imaging (MRI).” [Page 5, Lines 59]

1. Konno S, Hayashino Y, Fukuhara S, Kikuchi S, Kaneda K, Seichi A, et al. Development of a clinical diagnosis support tool to identify patients with lumbar spinal stenosis. Eur Spine J. 2007;16: 1951–1957. doi: 10.1007/s00586-007-0402-2.

2. Konno S, Kikuchi S, Tanaka Y, Yamazaki K, Shimada Y, Takei H, et al. A diagnostic support tool for lumbar spinal stenosis: a self-administered, self-reported history questionnaire. BMC Musculoskelet Disord. 2007;8: 102. doi: 10.1186/1471-2474-8-102.

3. Sugioka T, Hayashino Y, Konno S, Kikuchi S, Fukuhara S. Predictive value of self-reported patient information for the identification of lumbar spinal stenosis. Fam Pract. 2008;25: 237–244. doi: 10.1093/fampra/cmn031.

Line 66. Suggest to define NASS for readers here and remove from line 75.

Response: Thank you for this suggestion. We have defined NASS in the recommended sentence, and we have removed its definition from line 75. 

“The LSS-DST and LSS-SSHQ have been rated as having level II diagnostic evidence for LSS by the Degenerative LSS Work Group of the North American Spine Society (NASS) Evidence-Based Clinical Guideline Development Committee.” (Page 6, Line 69-70)

Line 80. Suggest to add “diagnostic guidelines” after NASS.

Response: Thank you for the suggestion. We have added “diagnostic guidelines” after NASS, according to your suggestion.

“Therefore, the superiority of the aforementioned two diagnostic support tools over the NASS diagnostic guidelines must be externally validated for situations in which a definitive diagnosis is made solely by an orthopedic surgeon” (Page 6, Line 82)

Line 90. I would appreciate more clarity on how this study data was collected. 1,657 hospitals is a large recruitment procedure. Was this data collected specifically for this study, or integrated into the standard data collection across these centers? Perhaps a national registry?

Response: Thank you very much for your question. This study was conducted as part of the lumbar spinal stenosis Diagnosis Support Tool (DISTO) project under the supervision of the Japanese Society for Spine Surgery and Related Research, to investigate the awareness and diagnostic accuracy of the LSS diagnostic support tool. We have added the following details about the data collection in our revised manuscript.

“This multicenter cross-sectional study used data from the Lumbar Spinal Stenosis Diagnostic Support Tool (DISTO) project, which was conducted from December 1, 2011 to December 31, 2012. The DISTO project was implemented in 1657 medical institutions under the guidance of the Japanese Society for Spine Surgery and Related Research (JSSR) to verify awareness and the diagnostic accuracy of a lumbar spinal stenosis diagnostic support tool in order to contribute to early detection and treatment of LSS. Recruitment for study participation was announced on the JSSR website, and the study was conducted at facilities that expressed a willingness to participate. An LSS-DST checklist and the NASS clinical description of LSS were distributed to participating medical facilities. The physician-in-charge completed the checklist, in addition to providing usual medical care. Patients who agreed to participate in the study were asked to complete the LSS-SSHQ prior to their consultation. The DISTO project collected and analyzed the checklist and the diagnostic information provided by the physician concerning LSS, peripheral artery disease (PAD), and diabetes mellitus (DM)” (Page 7, Line 93-Page 8, Line 105)

Line 181. Were those not included different from those that were? Is the data available to check this? It is important to know how these individuals differed in clinical characteristics. A selection bias may have been introduced where people with more severe disease were included, therefore inflating the diagnostic accuracy of the tools. For example, it appears that the age, sex, and presence of LSS could be compared between those included and those with missing or inappropriate data (n=7,338).

Response: Thank you for the suggestion. As you point out, if patients with missing values have mild disease, omitting these patients may introduce a bias in the severity of the target population and potentially overestimate the sensitivity of the screening tool. As suggested by the Reviewer, a comparison was made between participants included in the study and those with missing or inadequate data.

The results showed that participants included in the main analysis were older, that there was a higher proportion of men than women, and that the participants had a higher presence of LSS. This may suggest selection bias was introduced in terms of the participants included in the study. However, we included participants with a total score of >7 and those with a score of ≤3, despite missing ankle brachial index (ABI) and other values in sensitivity analysis. The number of participants in our sensitivity analysis was increased to 7,914, but the results were similar to those of the main analysis. Therefore, we consider the results of our study were robust.

We have added selection bias as a limitation of the study, and we have added a table comparing participants who were included and participants who were excluded as supporting information (S6 Table).

“Fifth, there is the possibility of selection bias because many participants were not included in the main analysis due to missing values. Therefore, we compared participants included in the main analysis with those who had earlier been excluded. Participants included in the main analysis were older, there was a higher proportion of men than women, and there was a higher presence of LSS (S6). Therefore, this may suggest that selection bias may be introduced in the participants included in the study. However, we included participants with a total score of >7 and those with a score of ≤3, despite lacking ABI and other values in sensitivity analysis. The number of participants in the sensitivity analysis was increased to 7,914, and the results were similar to those of the main analysis. Therefore, we consider the results of our study were robust.” (Page 28, Line 361 to Page 29, Line 370)

“S6. Comparison of the participants who were included and those who were excluded

Characteristic Participants who were included Participants who were Excluded P-value

 n(%) n(%) 

Age (years) 0.0008

20-29 166(5.0) 420(5.7) 

30-39 310(9.3) 780(10.6) 

40-40 308(9.3) 792(10.8) 

50-59 411(12.3) 976(13.3) 

60-69 892(26.8) 1824(24.9) 

70+ 1244(37.3) 2546(34.7) 

Sex <0.0001

Male 1755(52.9) 3576(48.8) 

Female 1564(47.1) 3756(51.2) 

Missing data 12 6 

Presence of LSS <0.0001

LSS(-) 1915(57.5) 5223(71.2) 

LSS(+) 1416(42.5) 2155(28.8) 

Each comparison was subjected to a χ-square test.

LSS, lumbar spinal stenosis

”

Figure 1. It is not clear how participants were excluded from the figure. I suggest that the “Excluded box” contain only those with missing data that were excluded. As written now, the numbers listed under exclusion sum to much greater than 7338. Please report missing data in a separate box. It would also be helpful to report how many patients were screened but declined to participate.

Response: Thank you for your suggestion. As you have suggested, we have divided Figure 1 into two parts and created a new figure (S1) with details of excluded cases because of missing data. Unfortunately, we did not collect data on patients who were screened but refused to participate, therefore we are not able to provide this information.

“Figure 1. Flow chart of participant inclusion

LBP, low back pain”

“S1. Details of excluded cases because of missing data

ABI, Ankle Brachial Index; DST, diagnosis support tool; LSS, lumbar spinal stenosis; NASS, North American Spine Society; SSHQ, self-administered, self-reported history questionnaire”

Line 183. It would be helpful to have more baseline characteristic information on all participants, since the diagnostic accuracy of a tool is dependent on the study population. Perhaps there was no other data collected, but if so, I would highly recommend greater description of the study cohort.

Response: Thank you for your important remarks. As you correctly indicate, we have not been able to collect any data other than those described in this study.

We have added the following details in our revised manuscript to describe the study cohort:

“This multicenter cross-sectional study used data from the Lumbar Spinal Stenosis Diagnostic Support Tool (DISTO) project, which was conducted from December 1, 2011 to December 31, 2012. The DISTO project was implemented in 1657 medical institutions under the guidance of the Japanese Society for Spine Surgery and Related Research (JSSR) to verify awareness and the diagnostic accuracy of a lumbar spinal stenosis diagnostic support tool in order to contribute to early detection and treatment of LSS. Recruitment for study participation was announced on the JSSR website, and the study was conducted at facilities that expressed a willingness to participate. An LSS-DST checklist and the NASS clinical description of LSS were distributed to participating medical facilities. The physician-in-charge completed the checklist, in addition to providing usual medical care. Patients who agreed to participate in the study were asked to complete the LSS-SSHQ prior to their consultation. The DISTO project collected and analyzed the checklist and the diagnostic information provided by the physician concerning LSS, peripheral artery disease (PAD), and diabetes mellitus (DM)” (Page 7, Line 93-Page 8, Line 105)

Lines 284-286. I am not convinced that real world diagnosis should be considered a strong gold-standard/criterion measure. I do agree with the authors in that specialist diagnosis is the best that is currently available, but I would like to see some discussion around the fact that a clinician-based reference standard test can be problematic. In fact, the authors do later discuss the limitations of having no consensus diagnostic definition for LSS, so I would suggest that the language around “our findings on diagnostic accuracy can be considered valid” be softened and some additional uncertainty be presented. It would also be prudent to comment on the potential variation in clinician diagnosis standards across the large number of participating clinicians. It is probable that different clinicians would arrive at conflicting results on the reference standard for LSS, at least in a proportion of patients.

Response: Thank you very much for your insightful comments. As you pointed out, the reference standard for this study is diagnosis by clinicians, which on itself is problematic. In addition to the fact that there is no international consensus on the definition of LSS, this is a relatively large-scale study, and there is a possibility that diagnoses may vary among different diagnosing clinicians. In light of the points you raised, we have changed the wording as follows.

“Second, the use of diagnosis by orthopedic surgeons as a reference standard reflects the best current diagnostic practice. Although the large number of facilities participating in the study may make standardization of clinicians' diagnostic procedures challenging, we consider that we were able to estimate the diagnostic accuracies of the screening tools at current standards of medical care.” (Page 27, Lines 336-340)

Line 299-301. It is a bold claim that the care setting is unlikely to alter the comparison of the three index tests. I do, however, respect the right of the authors to make this claim, but suggest that some reasoning be provided. Why would spectrum bias not be a concern here?

Response: Thank you for your very important point. We strongly agree that sensitivity values can vary depending on the research settings, but we considered that the difference in sensitivity when two screenings are applied to the same population would not be compromised. However, as you have pointed out, there is a possibility that spectrum bias may occur depending on the care setting, which may affect the comparison of diagnostic accuracy. Therefore, we have amended the sentences as follows:

“Further studies are warranted to confirm whether differences in diagnostic accuracy between the two diagnostic support tools and the NASS clinical descriptions estimated in this study apply to primary care settings.” (Page 28, Lines 353-355)

Reviewer #2

Reviewer #2: This study was conducted to compare the sensitivity and specificity of lumbar spinal stenosis diagnosis utilizing the Lumbar Spinal Stenosis-Diagnosis Support Tool (LSS-DST), the Lumbar Spinal Stenosis - Self-administered, Self-reported History Questionnaire (LSS-SSHQ) and the clinical description of LSS from the NASS diagnostic guidelines in a secondary care hospital setting. The authors concluded that “the LSS-DST and LSS-SSHQ had significantly higher sensitivity for diagnosing LSS in patients with LBP than the clinical description of LSS from the NASS diagnostic guidelines.” This reviewer has some criticisms, which should be addressed by the authors.

These tools have been used to “support” the diagnosis of LSS. This reviewer thinks that the most important thing in the management patients with LSS does prevent mis diagnosis when physicians, who are not spine specialists, use these tools in clinical setting. Considering the clinical relevancy, this reviewer thinks that it is important to precise if the score was not involved in the setting, the diagnosis of patients, who examined by the tools, must be “not” LSS. Therefore, the negative predictive value of these tools is more important than the sensitivity and specificity. Negative predictive values should be shown and compared among three tools.

Response: Thank you for your thought-provoking points. As you pointed out, the negative predictive value is also an important indicator for physicians who are not familiar with LSS examinations. The negative predictive value in this study can be calculated as follows:

NASS clinical description, 0.77 (95% CI 0.75 – 0.79); LSS-DST, 0.92 (95% CI 0.91 – 0.94); and LSS-SSHQ, 0.83 (95% CI 0.81 – 0.85).

The results also indicated that the LSS-DST and LSS-SHQ may be useful in ruling out the possibility of LSS compared with the clinical description of NASS. Therefore, we first listed the abbreviation for negative predictive value (NPV) on Page 12, Lines 163-167.

“We adopted a new cut-off point for the LSS-SSHQ (LSS-SSHQ version 1.1; a total score of 3 on Q1–Q4 or a score of ≥1 on Q1–Q4 and ≥2 on Q5–Q10 indicated positivity), as this cut-off point had higher sensitivity and negative predictive value (NPV) than the original value used in primary care settings [10].”

Next, in the “MATERIALS and METHODS” 2.5 Statistical analyses” sub-section (Page 15, Lines 200-202), the following sentence was added: 

“Furthermore, the NPVs of the three tools were also calculated, as it is important to determine the number of false positives obtained by physicians who were unskilled in examining LSS when using these tools clinically.”

In addition, the following information was added to the "RESULTS" section. (Page 21, Lines 258-264)

“The NPVs were 0.77 (95% CI 0.75–0.79) for the NASS clinical description, 0.92 (95% CI 0.91–0.94) for the LSS-DST, and 0.83 (95% CI 0.81–0.85) for the LSS-SSHQ (Table 5).”

Next, we added: 

Table 5. NPVs for the NASS clinical description of LSS, LSS-DST, and LSS-SSHQ

Index test NPVs

 Point estimate 95% CI

1) NASS clinical description of LSS 0.77 0.75–0.79

2) LSS-DST 0.92 0.91–0.94

3) LSS-SSHQ 0.83 0.81–0.85

CI, confidence interval; DST, diagnosis support tool; LSS, lumbar spinal stenosis; NASS, North American Spine Society; NPVs, negative predictive values; SSHQ, self-administered, self-reported history questionnaire” 

Since no universal analysis method has been established for NPV, no statistical tests for comparison have been performed.

Table 5 was changed to Table 6 (Page 22, Lines 270-274)

“Table 6. DORs of the NASS clinical description of LSS, the LSS-DST, and the LSS-SSHQ

Index test DOR

 Point estimate 95% CI

1. NASS clinical description of LSS 15.1 12.6–18.1

2. LSS-DST 33.3 26.9–41.1

3. LSS-SSHQ 7.0 5.9–8.3

CI, confidence interval; DORs, diagnostic odds ratios; DST, diagnosis support tool; LSS, lumbar spinal stenosis; NASS, North American Spine Society; SSHQ, self-administered, self-reported history questionnaire” 

Finally, in the "DISCUSSION" section, we have added the following description on Page 25, Lines 307-309:

“The high NPVs of the diagnostic support tools for LSS are important because of the clinical significance in terms of effectively limiting false-negative results, even for clinicians unfamiliar with diagnosing and treating patients with LSS.” 

Utilizing the typical clinical description of LSS according to the North American Spine Society (NASS), this reviewer does think that the description in point 1 is important items to diagnose the LSS. If this description was excluded, the NASS description to diagnose the LSS should not be used in the present study.

Response: As you have pointed out, the first of the NASS clinical descriptions, namely, degenerative LSS, describes a condition in which there is diminished space available for the neural and vascular elements in the lumbar spine secondary to degenerative changes in the spinal canal, is important for determining the morphological features of LSS. However, please note that the two diagnostic support tools in this study were designed to support the diagnosis of LSS based solely on clinical symptoms and physical findings, and do not include information obtained from radiological imaging studies. Therefore, like those tools, we treated the NASS as an index test and excluded the description on morphological features. Of course, when using the three descriptions of the NASS, it is clinically sensible to use it as a reference standard (i.e., the gold standard), not as an index test. 

Reviewer #3: 

This paper is well written. This study is a large-scale cross-sectional study and is noteworthy.

We agree that the two tools used in this study, LSS-DST and LSS-SSHQ can be used to improve the diagnostic accuracy of LSS.

Response: Thank you for your understanding of our research on the usefulness of LSS-DST and LSS-SSHQ. 

Reviewer #4

Reviewer #4: Q1. Why do authors hypothesize that LSS-DST and LSS-SSHQ would be more sensitive and more useful for screening than the clinical description of LSS from the NASS diagnostic guidelines? (Line 85-86)

Response: Thank you for pointing out that the LSS-DST and SSHQ are diagnostic tools that have undergone an appropriate development process and derivation. The NASS guidelines state that "the above definition of lumbar stenosis was developed by consensus after a global review of the literature and definitive texts," but these guidelines do not explicitly state the criteria by which the definition was ultimately determined. Therefore, since there is no guarantee that the definition will be defined to maximize sensitivity, we assumed that well-derived screening tools would be superior in terms of sensitivity.

Q2. Did the author use the list or questionnaire to confirm the clinical description of LSS from the NASS diagnostic guidelines? Or did the authors confirm the clinical description of LSS from the NASS diagnostic guidelines from the items of LSS-DST and LSS-SSHQ? Because at least from the previous reports from your group, the reviewer could not find the description regarding “the clinical description of LSS from the NASS diagnostic guidelines”.

Response: Thank you for your question. The questionnaire from the NASS diagnostic guidelines was used to confirm the clinical description of LSS.

Comment:

1. Authors emphasized sensitivity or screening for LSS. But authors also discuss the specificity, the reviewer thinks.

Response: Thank you for your suggestion. The lower sensitivity of the LSS-DST and LSS-SSHQ compared with the NASS clinical descriptions may be due to the inclusion of non-specific questions that apply to conditions other than LSS. For example, the LSS-DST adds two points just for being aged ≥71 years. Abnormal Achilles tendon reflexes are also often present in patients without LSS, especially in older adult patients and those with diabetes mellitus. With the LSS-SSHQ, numbness in the lower extremities is often present not only in patients with LSS, but also in patients with PAD and diabetes mellitus, as well as lumbar disc herniation. Urination during walking is often present in patients with an overactive bladder. Please note that our diagnostic support tools were developed for primary care physicians to screen for LSS and these tools value sensitivity over specificity to reduce the need for confirmatory testing by a specialist with detailed examinations and additional imaging studies.

2. I clinical setting, how do primary care physician use LSS-DST, LSS-SSHQ and the clinical description of LSS from the NASS diagnostic guidelines. Authors should describe it.

Response: Thank you for your suggestion. We have created the following description of how to use LSS-DST and LSS-SSHQ and have added it to the MATERIALS and METHODS section in sub-sections 2.3.1 and 2.3.2, respectively.

“2.3 Index tests

2.3.1 The LSS-DST

The LSS-DST is a brief clinical diagnostic tool that helps physicians precisely diagnose patients with LSS (Table 1) [3]. It consists of 10 items that are grouped into three main categories, namely, medical history, symptoms, and physical examination. The LSS-DST can be scored by primary care physicians within their usual resources without the need for special equipment or imaging studies.” (Page 10, Lines 128-131)

“2.3.2 The LSS-SSHQ

 The LSS-SSHQ was developed to evaluate the diagnostic value of the medical history of patients with LSS (Table 2) [4]. This self-completed questionnaire comprises 10 items concerning subjective symptoms only. The LSS-SSHQ can be distributed to patients by primary care physicians unfamiliar with neurological physical examination. Scoring can be completed by the patients or their primary care physicians.” (Page 12, Lines 160-163)

---

## [Editor Report · Decision Letter 1]

19 Apr 2022

Diagnostic accuracy of the Lumbar Spinal Stenosis-Diagnosis Support Tool and the Lumbar Spinal Stenosis-Self-administered, Self-reported History Questionnaire

PONE-D-21-38525R1

Dear Dr. Kurita,

We’re pleased to inform you that your manuscript has been judged scientifically suitable for publication and will be formally accepted for publication once it meets all outstanding technical requirements.

Kind regards,

Michael C Burger, M.D.

Academic Editor

PLOS ONE
---

## [Editor Report · Acceptance letter]

26 Apr 2022

PONE-D-21-38525R1 

Diagnostic accuracy of the Lumbar Spinal Stenosis-Diagnosis Support Tool and the Lumbar Spinal Stenosis-Self-administered, Self-reported History Questionnaire 

Dear Dr. Kurita:

I'm pleased to inform you that your manuscript has been deemed suitable for publication in PLOS ONE. Congratulations! Your manuscript is now with our production department. 

Kind regards, 

on behalf of

Dr. Michael C Burger 

Academic Editor

PLOS ONE